# Effect of Maternal Docosahexaenoic Acid (DHA) Supplementation on Offspring Neurodevelopment at 12 Months in India: A Randomized Controlled Trial

**DOI:** 10.3390/nu12103041

**Published:** 2020-10-03

**Authors:** Shweta Khandelwal, Dimple Kondal, Monica Chaudhry, Kamal Patil, Mallaiah Kenchaveeraiah Swamy, Deepa Metgud, Sandesh Jogalekar, Mahesh Kamate, Gauri Divan, Ruby Gupta, Dorairaj Prabhakaran, Nikhil Tandon, Usha Ramakrishnan, Aryeh D. Stein

**Affiliations:** 1Public Health Foundation of India, 47, Sector 44, Institutional area, Gurugram, Haryana 122003, India; dimple@ccdcindia.org (D.K.); monica.chaudhry@phfi.org (M.C.); ruby.gupta@phfi.org (R.G.); dprabhakaran@phfi.org (D.P.); 2Centre for Chronic Disease Control, C-1/52, 2nd Floor, Safdarjung Development Area, New Delhi 110016, India; 3KAHER’s JN Medical College, JNMC KLE University Campus, Nehru Nagar, Belgaum, Karnataka 590010, India; kamalpatil1967@yahoo.co.in (K.P.); mkswamy53@yahoo.co.in (M.K.S.); sandeshjogalekar555@gmail.com (S.J.); drmaheshkamate@gmail.com (M.K.); 4KAHER’s Institute of Physiotherapy, JNMC KLE University Campus, Nehru Nagar, Belgaum, Karnataka 590010, India; drdeepa_metgud@yahoo.com; 5Sangath, C-1/52, Block C 1, Bhim Nagri, Hauz Khas, New Delhi 110016, India; gauri.divan@sangath.in; 6Sangath Goa, H No 451 (168), Bhatkar Waddo, Socorro, Porvorium, Bardez, Goa 403501, India; 7All India Institute of Medical Sciences, Sri Aurobindo Marg, New Delhi 110029, India; nikhil_tandon@hotmail.com; 8Hubert Department of Global Health, Rollins School of Public Health, Emory University, Atlanta, GA 30322, USA; uramakr@emory.edu (U.R.); aryeh.stein@emory.edu (A.D.S.)

**Keywords:** maternal supplementation, pregnancy, lactation, docosahexaenoic acid (DHA), neurodevelopment, randomized controlled trial (RCT), India

## Abstract

Intake of dietary docosahexaenoic acid (DHA 22:6n-3) is very low among Indian pregnant women. Maternal supplementation during pregnancy and lactation may benefit offspring neurodevelopment. We conducted a double-blind, randomized, placebo-controlled trial to test the effectiveness of supplementing pregnant Indian women (singleton gestation) from ≤20 weeks through 6 months postpartum with 400 mg/d algal DHA compared to placebo on neurodevelopment of their offspring at 12 months. Of 3379 women screened, 1131 were found eligible; 957 were randomized. The primary outcome was infant neurodevelopment at 12 months, assessed using the Development Assessment Scale for Indian Infants (DASII). Both groups were well balanced on sociodemographic variables at baseline. More than 72% of women took >90% of their assigned treatment. Twenty-five serious adverse events (SAEs), none related to the intervention, (DHA group = 16; placebo = 9) were noted. Of 902 live births, 878 were followed up to 12 months; the DASII was administered to 863 infants. At 12 months, the mean development quotient (DQ) scores in the DHA and placebo groups were not statistically significant (96.6 ± 12.2 vs. 97.1 ± 13.0, *p* = 0.60). Supplementing mothers through pregnancy and lactation with 400 mg/d DHA did not impact offspring neurodevelopment at 12 months of age in this setting.

## 1. Introduction

The first 1000 days are crucial for a child’s neurodevelopment [1]. The brain develops rapidly through neurogenesis, axonal and dendritic growth, synaptogenesis, synaptic pruning, myelination, and gliogenesis [2]. These events build on each other, such that even small perturbations can have long-term effects on the brain’s structural and functional capacity [3]. Maternal nutrition during this time influences both pre- and postnatal growth, and development of the offspring [4,5,6].

It has been suggested that n-3 long-chain polyunsaturated fatty acids (LCPUFA) (especially docosahexaenoic acid [DHA]) levels enhance infant neurodevelopment [7,8,9,10,11]. The DHA is an important structural component of the human brain and retina. DHA accumulates in all of the brain regions and retinal photoreceptors [12]. These long-chain fatty acids regulate the fluidity of cell membranes as well as the activity of ion channels, enabling synaptic transmission and providing substrate binding to membrane receptors. The deposition of DHA in human brain phospholipids occurs primarily during the last trimester of pregnancy such as week 30 until the early postnatal periods continuing during the first two years of life [13,14,15]. Human fetuses and young infants have limited ability to synthesize n-3 LCPUFA de novo and are supplied via maternal (placental transfer, breast milk) or external (formula, dietary) sources [16,17]. Approximately 67 mg of DHA is accrued by the fetus per day [18]. Deprivation of n-3 LCPUFA, whether prenatally or after birth, has deleterious effects on learning abilities, memory, and visual grating acuity in monkeys, rats, and human infants [19,20].

DHA can be obtained from marine algae, fatty fish, and marine oils. Endogenous synthesis of DHA is limited, especially in the presence of excess n-6 LCPUFA. The Food and Agriculture Organization and World Health Organization (FAO/WHO) Expert Committee recommends a diet with a 5–10:1 ratio of n-6/n-3 LCPUFA and 300 mg/day of preformed DHA during pregnancy [21,22]. Since cereal-based diets are rich in n-6 LCPUFA but largely deficient in DHA-rich sources, population levels of plasma DHA are low in India. Studies report [23,24] that mean DHA intake was lowest (11 mg) among Indian pregnant women in the third trimester [25] compared to pregnant women from other developing countries like Bangladesh, Burkina Faso, Chile, China, India, and Mexico.

Studies conducted around the world to assess the association between intakes of DHA during pregnancy or lactation and neurodevelopmental outcomes in childhood have been inconsistent [7,26,27,28,29,30,31,32,33,34,35]. Few studies have assessed the effect of both prenatal and postnatal intake of DHA [36,37]. There is a paucity of data on the potential benefits of maternal DHA supplementation in infants, especially in the Indian population. The present study examines the hypothesis whether 400 mg/d maternal DHA supplementation from ≤20 weeks through 6 months postpartum influences infant neurodevelopment at 12 months of age. The present study DHANI (Maternal **DHA** supplementation and offspring Neurodevelopment in India) is the first to examine the effects of maternal DHA supplementation from mid-pregnancy through six months postpartum on postnatal neurodevelopment in India.

## 2. Materials and Methods

### 2.1. Study Design

DHANI was a randomized, double-blinded, placebo-controlled trial to test the effect of providing pregnant women 400 mg/d algal DHA compared to placebo from ≤20 weeks of singleton gestation through 6 months postpartum on offspring neurodevelopment. The trial protocol has been published elsewhere [38]. The protocol was reviewed and approved by institutional review boards (IRBs) of all participating institutions: Center for Chronic Disease Control (CCDC-IEC_04_2015), Public Health Foundation of India (TRC-IEC-261/15), Jawaharlal Nehru Medical College (MDC/IECHSR/2016-17/A-85), and All India Institute of Medical Sciences (IEC-28/17.11.2015).

### 2.2. Participants

Healthy pregnant women (18–35 years; ≤20 weeks single gestation; with no medical complications or chronic diseases) attending the Department of Obstetrics and Gynecology at the Prabhakar Kore hospital (PKH) in a large city in southwest India for their prenatal check-ups were approached by designated project staff. The consulting obstetrician on site, considering obstetric history and complications, affirmed final eligibility. After signing a witnessed informed consent form in their preferred local language (Kannada, Marathi, or Hindi), consenting women (n = 957) were randomized to receive either 400 mg of DHA or a placebo until six months postpartum as explained later. Baseline assessments at enrollment included sociodemographic characteristics, dietary intake, obstetric history, anthropometric measurements, blood investigations including a non-fasting blood draw, vital signs. The women received their first 15-day supply of supplements in the form of coded bottles. Further supplements were provided at the women’s homes every fortnight by fieldworkers.

The next study hospital visit was at delivery (referred to as E0). A research officer stayed in close contact with each mother after she entered her last trimester and visited the mother within 24 h of delivery to collect information on the type of delivery, complications if any, and newborn anthropometry, and obtained a breast milk sample. We also collected and stored maternal and cord blood samples from the enrolled mother–child dyads.

Postpartum visits for mother–child dyads at 1 month (E1), at 6 months (E6), and at 12 months (E12) postpartum were scheduled. E1 data collection (neonatal anthropometry, maternal anthropometry, breastfeeding pattern, breast milk sample (transported to site in ice box), overall health of both mother and child, etc.) was carried out at home by trained fieldworkers. Infant neurodevelopment was assessed at the hospital at E6 and E12 by trained and certified psychologists.

A pre-piloted and validated semi-quantitative food frequency questionnaire (FFQ) with n-3 LCPUFA-rich Indian foods was administered to all women at enrollment. Dietary data were also collected in a subsample of women (n = 278) using a 24-h diet recall within 1 month of recruitment. Nutrient intakes were calculated using DIETSOFT software (http://dietsoft.in/) based on standardized nutritive values of Indian foods [39]. A study physician reviewed adverse events.

### 2.3. Randomization and Masking

A computer-generated randomization using a permuted block design was used to generate the randomization list. The code list was generated for 1200 women to allow for potential loss to follow up. The assignment code list was placed in a sealed envelope at the beginning of the study and was held in a sealed location by a staff member at Gurugram, Public Health Foundation of India (PHFI). This list was used by this person to pre-code the supplement bottles in the warehouse before the bottles arrived on site, ready for distribution. All study participants and members of the study team (including members at the study site) remained blinded to the treatment allocation until the analysis of all data collected was carried out. Blinded preliminary descriptive analyses were presented to Data Safety Monitoring Board (DSMB) and once approved, full statistical analyses were undertaken. Unblinding occurred only after primary tables were generated.

The intervention was either 400 mg/d algal DHA or a matching placebo (soy/corn oil) delivered in the form of similar-looking softgel capsules, donated by DSM Nutritional Supplements, Mumbai. The composition of all capsules was same except for the DHA. Enrolled women received the intervention from the date of randomization until six months postpartum.

The shelf life of the capsules at room temperature (25 °C) was 24 months from the date of manufacture (90 days once the bottle was opened). The coded capsule bottles were stored in an on-site refrigerator for extra safety to slow down oxidation. Each bottle contained a fortnightly supply of capsules. Bottles were distributed by fieldworkers during scheduled home visits. The women were instructed to take two capsules (each with 200 mg DHA or placebo) daily, preferably at the same time each day. They were told to keep them in a cool, dry place. Supplements were provided for more than two weeks in cases where the woman shared plans to travel.

### 2.4. Procedures

Subjects were asked to maintain a daily record in an easy-to-fill log (piloted and found suitable for population with low literacy levels) of their supplement consumption. Weekly calls were made to encourage compliance and inquire about the general well-being of the women. The used bottles were collected (for pill count) by the fieldworkers during the fortnightly home visits. Red blood cell (RBC) phospholipids were analyzed for DHA levels. The analyses of breast milk samples have not been presented here.

### 2.5. Outcomes

Development quotient (DQ) as a marker of neurodevelopment, among infants at 12 months of age, was assessed by a trained psychologist using the Developmental Assessment Scale for Indian Infants (DASII). The DASII tool is the Indian modification of the Bayley Scales of Infant Development (BSID), using Indian norms for 67 motor and 163 mental items. DASII provides a measure of development for Indian infants below 30 months of age. DQ is defined as the ratio of functional to chronological age. Third, 50th and 97th percentile norms have been generated for Indian children (DQ range of 35–160 ± 3.5). The maximum DQ score is 160. In terms of interpretation of scores, DQ score ≥85 is normal and 71–84 is mild to moderate delay. Severe developmental delay is defined on DASII as DQ score ≤70 (≤2 SD). The inter-correlation between the motor and mental performance on the two sections of the scale ranges from 0.24 to 0.62 [40]. The motor development items cover the child’s development from supine to erect posture, neck control, locomotion, and manipulative behavior such as reaching, picking up, handling things, and so forth. The mental development items record the child’s cognizance of objects in the surroundings, perceptual pursuit of moving objects, development of communication and language comprehension, spatial relationship and manual dexterity, imitative behavior, social interaction, and so forth.

### 2.6. Biomarkers

Fatty acid composition of plasma phospholipids and red blood cell (RBC) membrane phospholipids are appropriate biomarkers of fatty acid status and related to dietary intakes [41,42]. Since the plasma (reflecting short-term intake) and RBC (reflecting long-term intake) fatty acid composition are related, the markers of longer-term intake RBC phospholipid fatty acids (linoleic acid (18:2n-6); alpha-linolenic acid (18:3n-3); arachidonic acid (20:4n-6); eicosapentanoic acid (20:5n-3); DHA (22:6n-3)) were measured. Lipids were extracted from RBCs using the method of Rose and Ocklander [43], phospholipids were separated by thin-layer chromatography. The phospholipids were trans-esterified using the method by Lepage and Roy [44] and fatty acids were identified by gas chromatography with flame ionization detector. Thirty-seven fatty acid methyl ester mix from Supelco (SIGMA-ALDRICH) was used to identify the fatty acids using their retention time.

Maternal nonfasting blood samples (5 mL) were obtained by venipuncture at recruitment, delivery, and six months postpartum. Neonatal blood samples were obtained from the umbilical cord vein immediately after delivery using the syringe method. A 2 mL venous blood sample was obtained from infants at 6 and 12 months of age. All samples were collected into tubes containing disodium ethylene diamine tetra acetic acid (EDTA). Plasma was separated the same day by cold centrifugation (4 °C) at 800× *g* for 10 min and RBCs were washed thrice using equal volumes of saline. Plasma and washed RBCs were stored at −80 °C for later analysis.

Breast milk samples (one day, one month, and six months postpartum) were collected. The milk samples were from a morning feed but not the first one, preferably between 08:00 am and 12:00 pm. Infants were allowed to suckle the nipple for a few minutes, and then a breast milk sample (10 mL) was expressed manually by the mother herself, after which the feeding continued. The samples were refrigerated immediately to prevent bacterial growth and later aliquoted into smaller 2 mL containers, filled nearly to the top to minimize oxidation, and frozen at −80 °C until analysis. Although the amount of fat may change within and between feeds, the proportion of fatty acid remains relatively constant. Since breast milk PUFA were being expressed as a percent of total fatty acids, complete breast expressions were not required [45]. RBC phospholipids were analyzed by gas chromatography using standard methods.

### 2.7. Statistical Analysis

The primary study outcome was infant development quotient (DQ) score (composite score of motor and mental development) obtained at 12 months of age. We required 674 mother–child pairs to detect an effect size of 0.25 with 90% power [46].

Maternal household and offspring characteristics in the two treatment groups were summarized to assess the effectiveness of randomization. Continuous, normally distributed variables were summarized as means and standard deviations, while skewed variables were summarized as medians and interquartile ranges. Categorical/binary variables were summarized using proportions. Analysis was done using the intent to treat (ITT) principle. Per protocol analyses were also carried out including infants who had a valid 12-month DQ score (DASII test administered at 12 months postpartum ± 4 weeks) and whose mothers’ compliance rate >80%. All protocol deviation cases were excluded from per protocol analyses. We compared baseline characteristics between the final study sample and those who were lost to follow-up.

For the primary outcome, we used a two-sample t test to compare the mean DQ (mental and motor) score at 12 months between the DHA and placebo group using DASII. We performed the subgroup analysis using a two-sample *t*-test for the difference in mean DQ score between DHA and placebo group stratified by pre-specified subgroups, that is, mother’s age (18–20, 21–25, 26–30, 31–35 years), mother’s BMI (<18, 18.0–22.9, 23.0–24.9, ≥25 kg/m^2^), gravidity (multigravida, primigravida), gestational age at delivery (<37, ≥37 weeks), duration of supplementation (≤10, 10.1–12.0, ≥12.1 months), vegetarian diet (yes, no), physical activity (inactive/low, moderate, high), and gender of the child (male, female). The *p*-value for interaction was calculated by including the interaction term between the characteristic of interest and treatment group in the linear regression model. Heterogeneity was assessed based on the significance of the interaction term between the characteristic of interest and treatment (*p* < 0.05). All the statistical analysis was done using STATA 16.0 version (College Station, TX, USA) and R 3.6.2 version.

## 3. Results

Enrollment began in January 2016 and ended in August 2017. Figure 1 shows the CONSORT flow chart. We screened 3379 women for eligibility and found 1131 to be eligible. A total of 957 mothers were enrolled and randomized (DHA = 478; Placebo = 479). Of the 902 live births, 878 were followed to 12 months and the DASII was administered to 863 of those. Loss to follow-up (8.2% from randomization through 12 months) and mean ± SD compliance did not differ by group (DHA: 93.0 ± 10.1; Placebo: 92.7 ± 10.7). Baseline maternal and household characteristics were similar by treatment group (Table 1). Overall, the mean age of the mothers was 23.9 (3.6) years, the mean BMI of the participating women was 20.6 ± 3.6 kg/m^2^, and the mean hemoglobin was 11.1 ± 1.3 gm/L. Based on the semiquantitative FFQ administered at baseline, the dietary intake of n-3-rich foods was similar across the two groups. Additionally, there was no significant difference in the baseline characteristics between those who completed the study and those who did not complete the study (Appendix A).

The proportion of live births was similar across both the groups (DHA vs. Placebo: 94.1% vs. 94.3%) (Table 2). Initiation of breastfeeding in 1 h did not differ across groups (DHA vs. Placebo: 79.9% vs. 81.7%) (Table 3).

The age at which complementary feeding started was similar across groups (DHA vs. Placebo, mean ± SD: 5.4 ± 0.67 vs. 5.3 ± 0.83 months) (Table 3).

The maternal RBC DHA concentrations (mol % of fatty acid) (mean ± SD) were not different between the groups at baseline (DHA; 0.86 ± 0.78, Placebo: 0.88 ± 0.71) but significantly higher in the DHA group compared to placebo group at delivery (DHA: 1.94 ± 1.42, Placebo: 0.84 ± 0.56; *p* < 0.001) (Table 4).

In the placebo group, there was no change in RBC DHA concentrations over the period of intervention while in the DHA group, concentrations increased by 1.04 (0.86, 1.23) mol % of fatty acid, paired *t*-test, *p* < 0.001.

The DQ score (mean ± SD) at 12 months for the DHA and placebo groups was 96.6 ± 12.2 and 97.1 ± 13.0, respectively, and the mean difference (placebo minus DHA) was 0.46 (95% CI: −1.23, 2.14; *p* = 0.60). The motor and mental scores also did not differ significantly across the groups at 12 months (Table 5). There were 25 serious adverse events, all determined to be unrelated to intervention (DHA group = 16; placebo = 9; *p* = 0.154) (Appendix A). The last 12-month assessment (including DASII) was conducted in early April 2019.

The per protocol analysis also showed no difference in 12-month DQ scores between the two groups (Appendix A). The mean infant DQ, mental and motor scores at six months did not differ significantly across groups (secondary outcomes not shown in this paper). The mean DQ score did not differ significantly across mother’s age group, BMI, gravida, gestation age at delivery, compliance rate, duration of supplementation, physical activity (Figure 2).

However, the mean DQ score for female child was higher in placebo as compared to DHA [mean difference (placebo− DHA) (95% CI): 2.74 (0.40, 5.09)]. The p-value for interaction was significant for gender of the child (*p* = 0.013). The pre- and postnatal maternal supplementation with 400 mg/d DHA did not improve the infant’s DQ score as measured by DASII at 12 months of age.

## 4. Discussion

In this well designed and executed RCT (CONSORT checklist enclosed as Appendix A), pre- and postnatal maternal supplementation with 400 mg/d DHA did not improve the infant’s development score as measured by DASII at 12 months of age. Our findings are in accordance with some other trials [46,47,48,49]. The reviews published so far in this field to understand the association between maternal DHA supplementation and infant neurodevelopment have also reported the evidence of a positive association between maternal DHA supplementation and infant neurodevelopment to be either too low or inconsistent, with a majority of the RCTs showing no positive effect [36]. Makrides et al. provided 800 mg/d and [50] showed no difference in children’s cognitive scores between the intervention and control group at 18 months. Similarly, Helland et al. [49] found no effect of prenatal supplementation with cod liver oil on cognitive development among 288 three-month-old children in Norway. On the other hand, the randomized trial of Colombo et al. [51] found that lower doses of DHA supplementation of the infants (4 to 9 months of age) showed better cognitive outcomes in terms of their attention span. Studies have reported maternal DHA status in pregnancy to be positively associated with infant’s brain volume at 1 month [52], improved infant’s attention [53], and enhanced problem-solving skills at 12 months [54]. Rees et al. [26] showed that the infants of mothers who had a higher dietary DHA intake during the second and third trimesters of their pregnancy performed better on cognitive assessment measures (habituation and sustained visual attention) at 4.5 and 9 months of age. However, it was also observed that the infants from the medium dietary DHA group [0.64% of fatty acids from DHA (34 mg/100 kcal)] performed significantly better than the low-DHA [0.32% (17 mg/100 kcal)] and high-DHA group [0.96% (51 mg/100 kcal)], with the latter showing the worst performance [26,51]. Additionally, some studies have reported improved cognition among older children whose mothers were supplemented with 400–600 mg/d DHA prenatally, after they turned four years old [27,30,35].

The neurodevelopment in the current study was assessed with the help of the DASII test which is BSID’s adaptation. Although BSID is a global standardized test for assessing cognitive functioning and has been frequently utilized in neurodevelopment effects of LCPUFA in infants [55], its sensitivity to pick up some subtle differences in infant cognitive ability has been questioned [56]. Experts suggest that differences in intellectual functioning that are sensitive to pathways influenced by n-3 fatty acids may only be detected with more sensitive measures of neurodevelopment such as neuroimaging techniques [57], neuropsychological testing [58], distinct cognitive abilities, or executive functioning [59]. The suitability of these approaches for large field-based trials in low resource settings warrants further exploration.

Few possible limitations of our single-center trial may include lack of details on home environment and detailed dietary data on the children which may have influenced the neurodevelopment in the first year of life. The measure of neurodevelopment used in the study (i.e., the DQ score) has not been validated against the functional magnetic resonance imaging (fMRI) which is currently considered the gold standard noninvasive hemodynamic-based neuroimaging technology. The fatty acid composition of the blood and breast milk collected from the enrolled women are not currently available. We also did not estimate dietary n-3 LCPUFA intakes in all the study subjects; however, the biomarker (RBC phospholipid DHA) is very responsive to intake and was measured at both enrollment and immediately after birth as an indicator of prior status and study DHA intake. Further, other fatty acids like AA and DHA:AA ratios may play a key role in neurodevelopment [60]. To the best of our knowledge, our study is the first one to examine the effects of in utero and early-life DHA exposure (through maternal supplementation from mid-pregnancy through six months postpartum) on postnatal neurodevelopment of Indian infants. Through a long supplementation phase and follow-up period, the participants of our trial reported good compliance and very low attrition. Genetic predisposition, prenatal and postnatal care and nutrition, and social and physical environment may all be critical in shaping the neurodevelopment of an infant. We collected maternal dietary data at baseline but not at later visits. Changes throughout the pregnancy with respect to not only n-3 LCPUFA but also other nutrients such as vitamin D or iron [61,62] may affect offspring neurodevelopment.

While DHA is a key intrinsic factor constituting more than 40% of brain polyunsaturated fatty acids [63,64], external factors like care and stimulation in the home environment also play a significant role in a child’s cognitive development [65,66]. Poor physical conditions of home and limited access to age-appropriate learning materials have been linked with social–emotional problems in children. Ramakrishnan et al. [46] indicated a possible attenuating effect of DHA on the positive association between home environment and psychomotor development. DHA may be especially helpful for children living in home environments characterized by reduced caregiver interactions and opportunities for early childhood stimulation [67]. It might have been useful to have these data in the present study. Further, the mean DQ score of girl children in the placebo group was higher. This inconsistency may be due to chance. Some studies also attribute gender-related differences in neurodevelopment to sex hormones [68] and/or social context [69], but since these were not assessed in our trial, we cannot be certain of this.

In summary, supplementing mothers through pregnancy and lactation with 400 mg/d DHA (vs. placebo) did not benefit offspring neurodevelopment at one year of age in this Indian setting. Deeper insights into maternal dietary patterns, young child feeding practices, home environment, and the interactions amongst these factors are warranted to understand what shapes early neurodevelopment.

## Figures and Tables

**Figure 1 nutrients-12-03041-f001:**
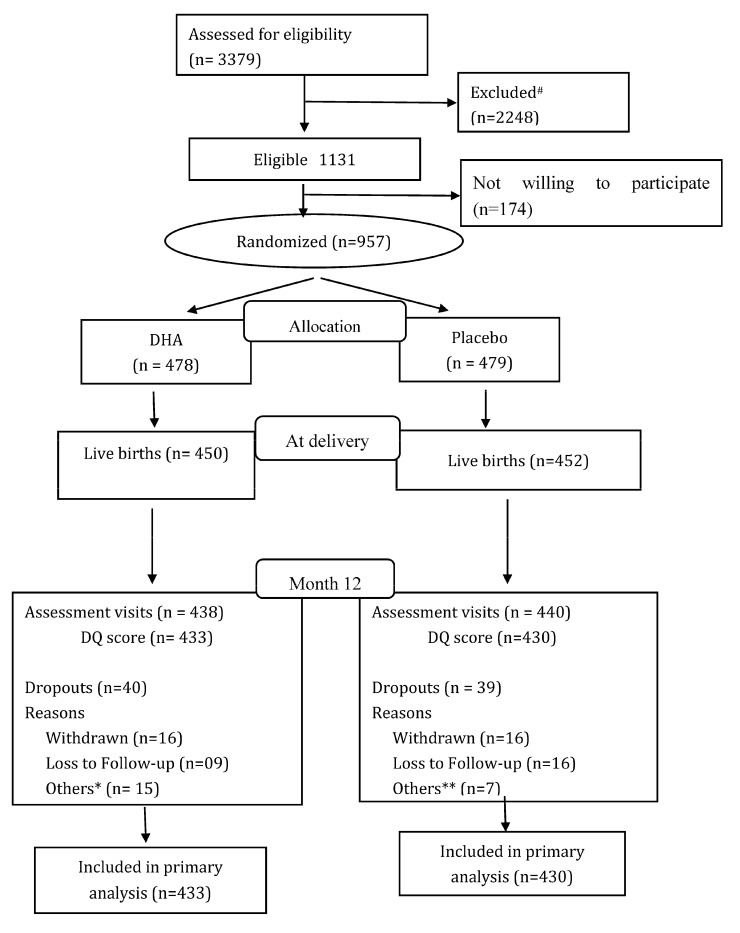
Consort. ^#^ Reasons for exclusion: gestational diabetes (n = 69); Hb < 7 gm% (n = 46); gestational age >20 weeks (n = 673); high-risk pregnancies (n = 118); chronic conditions (n = 246); under any other trial (n = 4); delivery plan other than PK (n = 835); missing/wrong contact information (n = 257). * Others included: abortion (n = 1); abruptio placenta (n = 1); fresh stillbirth (n = 4); macerated stillbirth (n = 3); neonatal death (n = 2); maternal death (n = 1); congenital anomalies (n = 1); infant death (n = 2) in DHA group. ** Others included: fresh stillbirth (n = 4); macerated stillbirth (n = 2); medical termination (n = 1) in Placebo group.

**Figure 2 nutrients-12-03041-f002:**
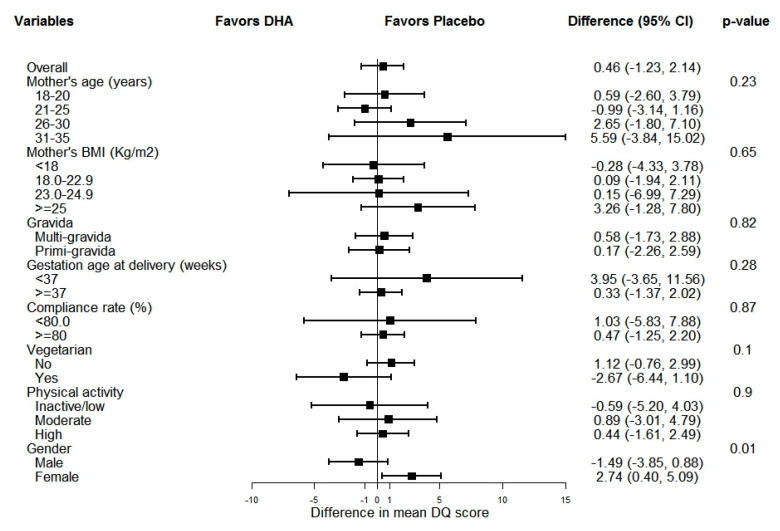
Subgroup analysis. Difference: Placebo minus DHA; Difference in mean DQ score between DHA and placebo group at 12 months was calculated using two-sample *t*-test for each subgroup; *p*-value for interaction calculated using linear regression model including interaction term for characteristic of interest and treatment group.

**Table 1 nutrients-12-03041-t001:** Baseline Characteristics.

	DHA (*N* = 478)	Placebo (*N* = 479)
Maternal age (years), mean ± SD	23.5 ± 3.5	23.6 ± 3.7
Gestational age at enrollment (weeks), median (p25, p75)	15.0 (12, 18)	15.0 (12, 18)
Primigravida, n (%)	180 (37.7%)	206 (43.0%)
Education, n (%)		
College graduated and above	88 (18.4%)	82 (17.1%)
High school/secondary	371 (77.6%)	386 (80.6%)
Employed, n (%)	119 (25.0%)	104 (22.0%)
Household income (>Rs 20,000), n (%)	65 (13.6%)	47 (9.8%)
Dietary habits—vegetarian, n (%)	73 (15.3%)	87 (18.2%)
Consuming fish/seafood, n (%)	258 (53.9%)	202 (57.8%)
Anthropometric measurements		
Height (cm), mean ± SD	154.1 ± 5.6	153.9 ± 5.7
Weight (kg), mean ± SD	48.9 ± 9.0	48.9 ± 8.5
BMI (kg/m^2^), mean ± SD	20.5 ± 3.5	20.7 ± 3.6
MUAC, (cm), mean ± SD	24.3 ± 3.0	24.3 ± 3.1
Biochemical measures		
Hb (gm%), mean ± SD	11.1 ± 1.3	11.2 ± 1.3
DHA (mol % of fatty acid) *-, mean ± SD	0.86 ± 0.78	0.88 ± 0.71

BMI: Body mass index; MUAC: Mid upper arm circumference; Hb: Hemoglobin; DHA: Docosahexaenoic acid; *- *N* = 258 (DHA); *N* = 224 (Placebo). Data are presented as mean ± standard deviation or median (p25, p75) or number (%).

**Table 2 nutrients-12-03041-t002:** Offspring characteristics at delivery.

At Delivery	DHA *N* = 450	Placebo *N* = 452
Live births, n (%)	450 (94.1%)	452 (94.3%)
Gestational age at the time of delivery (weeks), median (p25, p 75)	39.0 (38.0, 40.0)	39.0 (38.0, 40.0)
Delivery place—study center, n (%)	372 (84.7%)	375 (85.2%)
Delivery conducted by doctor, n (%)	416 (94.5%)	423 (96.1%)
Spontaneous labor, n (%)	410 (93.2%)	407 (92.5%)
Caesarean, n (%)	154 (35.0%)	175 (39.8%)
Male child, n (%)	234 (52.0%)	243 (54.0%)

Data are presented as median (p25, p75) or number (%); denominator for live births DHA (*N* = 478); Placebo (*N* = 479).

**Table 3 nutrients-12-03041-t003:** Feeding practices.

	DHA n/N (%)	Placebo *n*/*N* (%)
Initiation of breastfeeding in 1 h, n (%)	346/433 (79.9%)	357/437 (81.7%)
Exclusively breastfed until 6 months, n (%)	236/426 (55.4%)	237/421 (56.3%)
Age at which complementary feeding initiated (months) *, mean ± SD	5.4 ± 0.67	5.3 ± 0.83

Data are presented as number (%); * DHA (*N* = 166); Placebo (*N* = 156).

**Table 4 nutrients-12-03041-t004:** Mean DHA (mol % of fatty acid) levels in RBC phospholipids.

	DHA	Placebo	Mean Difference * [95% CI]	*p*-Value
DHA at baseline	*N* = 258 0.86 ± 0.78, 0.56 (0.32, 1.21)	*N* = 224 0.88 ± 0.71, 0.55 (0.37, 1.28)	0.02 [0.11,0.15]	0.77
DHA at the time of delivery	N = 271 2.03 ± 1.75, 1.41 (0.61, 2.99)	N = 242 1.12 ± 0.86, 0.83 (0.42, 1.72)	−0.91 [−1.16, 0.67]	<0.001
DHA in cord blood	N = 265 2.61 ± 1.45, 2.64 (1.38, 3.81)	N = 232 1.83 ± 0.90, 1.76 (1.16, 2.48)	−0.77 [−0.99, 0.56]	<0.001
DHA in infant blood at 6 months	N = 263 1.94 ± 1.42, 1.65 (0.64, 3.08)	N = 240 0.84 ± 0.56, 0.77 (0.43, 1.12)	−1.09 [−1.29, −0.9]	<0.001
DHA in infant blood at 12 months	N = 227 1.71 ± 1.18, 1.50 (0.60, 2.64)	N = 204 1.27 ± 0.93, 1.03 (0.43, 1.89)	−0.44 [−0.64, −0.24]	<0.001

Data are presented as mean ± standard deviation, median (p25, p75); DHA: Docosahexaenoic acid; *p*-value calculated using unpaired *t*-test; * difference = Placebo minus DHA.

**Table 5 nutrients-12-03041-t005:** DQ score and components of DQ (motor and mental scores) at 12th month.

At 12th Month	DHA (n = 433)	Placebo (N = 430)	Difference * [95% CI]	*p*-Value
DQ score	96.6 ± 12.1	97.1 ± 13.0	0.46 [−1.23,2.14]	0.60
Motor score	47.6 ± 3.7	47.6 ± 3.7	0.03 [−0.47,0.52]	0.92
Mental score	106.0 ± 7.0	106.7 ± 7.6	0.63 [−0.35,1.61]	0.21

Data are presented as mean ± standard deviation and [95% CI]; unadjusted model: *t*-test for difference between mean values at 12 months; DQ: development quotient. * Placebo minus DHA.

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
