# Peer review of "Effect of Maternal Docosahexaenoic Acid (DHA) Supplementation on Offspring Neurodevelopment at 12 Months in India: A Randomized Controlled Trial"

_nutrients, 2020, doi:10.3390/nu12103041_

Round 1

Reviewer 1 Report

I would like to thank the authors for the enormous effort they put into this protocol and writing this manuscript. I think it is a very interesting article and it covers many of the strengths of research in LCPUFA and neonatal health. However, there are many limitations to the study and that needs more work. In the first place, the article is not in accordance with the template or the author's guidelines of Nutrients. On the other hand, the hypothesis and objectives are not defined. There are certain gaps in the description of recruitment. The statistical analyses require a deep revision. More details can be found in the attached file. 

Thanks

Reviewer 2 Report

This RCT is designed very well and follow the CONSORT statement.  The introduction is adequate and produces sufficient scientific background to the topic of the paper. The main weakness of your work is the biomarker study method (see comment 1). The results are not clearly reported. eTable 2 is not immediately understandable, please consider converting in a graphic format and insert in the main paper, instead of the supplementary material. The conclusions are well supported by the study's results.

Reviewer's comments list:

1- Line 191, Plasma and RBC fatty acid composition are not so well related. Erythrocytes better reflect long-term dietary fatty

acid intakes than does plasma or serum, that respond more quickly to dietary modification of previous days. Please see this paper:

 "Am J Clin Nutr 2007;86:74–81".

2- Line 196, Please detailed the motivation of non-fasting blood samples employed. The fatty acid analysis is affected by food ingestion in the prior 12 hours (e.g. chylomicrons).

3- Line 201 Please, insert specific temperature of cold centrifugation (e.g. 4° C).

4- Line 204 Please, insert a colon between numbers indicating hours (08:00 and 12:00).

5- Line 212 Please, detailed standard methods of GC analysis.

6- Line 280, Please change the term "serum" with "plasma", because you have analysed plasma, not serum.

Author Response

REVIEWER 2

1.    Line 191, Plasma and RBC fatty acid composition are not so well related. Erythrocytes better reflect long-term dietary fatty acid intakes than does plasma or serum, that respond more quickly to dietary modification of previous days. Please see this paper:

 "Am J Clin Nutr 2007;86:74–81".

2.    Line 196, Please detailed the motivation of non-fasting blood samples employed. The fatty acid analysis is affected by food ingestion in the prior 12 hours (e.g. chylomicrons).

Since analysis has been done using RBCs, no chance of interference with chylomicrons etc.

3.    Line 201 Please, insert specific temperature of cold centrifugation (e.g. 4° C).

Correction made

4.    Line 204 Please, insert a colon between numbers indicating hours (08:00 and 12:00).

Corrections made in manuscript

5.    Line 212 Please, detailed standard methods of GC analysis.

The paragraph is added in Biomarkers section:

Lipids were extracted from RBCs using method of Rose & Ocklander (1965),  phospholipids were separated by thin layer chromatography .  The phospholipids were transesterified using method by Lepage and Roy [43] and fatty acids were identified by gas chromatography with flame ionization detector. 37-fatty acid methyl ester mix from Supelco (SIGMA-ALDRICH) was used to identify the fatty acids using their retention time.

6.    Line 280, Please change the term "serum" with "plasma", because you have analysed plasma, not serum.

RBCs used for analysis. Correction made in manuscript.

Round 2

Reviewer 1 Report

Thank you for incorporating the changes and addressing the questions. However, I believe that there are still a few things that need to be improved. I have two major comments and a few minor comments that need to be addressed in the manuscript.

Major comments:

  1. Figure 1 should be easy to follow, simple, and clear as it is difficult to understand. Some of the information can be mentioned in a footnote to make it clean and clear. The following link can be used for the template. (PMID: 32298381)

https://pubmed.ncbi.nlm.nih.gov/32298381/#&gid=article-figures&pid=fig-1-uid-0

  1. Figure 2, still it is not clear what statistical test has been performed to get the p-value? Is the p values for interaction term? If yes, what analysis/model was performed to get the p-value? Please mention in detail what statistical analysis has been performed to get the mean difference and p-value shown in Figure 2.
  2. What does the author think about the gender effect in Figure 2? What does it mean in terms of its relevance and how the author can explain this gender effect and its significance?

Minor comments:

  1. Table 4 can be improved by merging two rows in one (except for the first row).
  2. Why line 306-308 is in italics? Please correct.
  3. Table 4 title is not is bold but for others it is bold. Please correct and be consistent throughout.
  4. In Table 5 the first row in bold while in other it is not. Please be consistent and correct.
  5. Please make sure to follow the guideline for Nutrients for the manuscript.
  6. At line 360 it should be center. Please correct.
  7. Please write a p-value instead of p value.

Thank you

Author Response

Dear Reviewer 1,

We are very grateful for the meticulous reviewing of our manuscript. We hope we have been able to address all comments properly. Please do let us know if you think we can improve any aspect further. 

Best Regards

Shweta
